# Protocol for a feasibility randomised controlled trial of the 'Outdoor' mobility intervention for older adults after hip fracture

Katie J. Sheehan[1,2]*, Denise Bastas[2], Stefanny Guerra[1], Siobhan Creanor[3], Claire Hulme[3], Sallie Lamb[4], Finbarr C. Martin[2], Catherine Sackley[5], Toby Smith[6], Philip Bell[7], Melvyn Hillsdon[8], Sarah Pope[9], Heather Cook[3], Emma Godfrey[2]

1 Bone & Joint Health, Blizard Institute, Queen Mary University of London, London, United Kingdom, 2 Department of Population Health Sciences, School of Life Course and Population Health, Faculty of Life Science and Medicine, King's College London, London, United Kingdom, 3 Department of Health and Community Sciences, Medical School, Faculty of Health and Life Sciences, University of Exeter, Exeter, United Kingdom, 4 Department of Public Health and Sport Sciences, Medical School, Faculty of Health and Life Sciences, University of Exeter, Exeter, United Kingdom, 5 School of Health Sciences, Queens Medical Centre, University of Nottingham, Nottingham, United Kingdom, 6 Warwick Medical School, University of Warwick, Coventry, United Kingdom, 7 Public and Patient Involvement Member Representative from Trauma Rehabilitation (Orthopaedic) for Older People (TROOP), London, United Kingdom, 8 Public Health and Sports Sciences, Medical School, Faculty of Health and Life Sciences, University of Exeter, Exeter, United Kingdom, 9 Integrated Falls & Bone Health Service, St John's Therapy Centre, St Georges University Hospital NHS Foundation Trust, London, United Kingdom

* k.sheehan@qmul.ac.uk, katie.sheehan@kcl.ac.uk

## Abstract

### Background

A high proportion of patients do not regain outdoor mobility after hip fracture. Rehabilitation explicitly targeting outdoor mobility is needed to enable these older adults to recover activities which they value most. The overarching aim of this study is to determine the feasibility of a randomised controlled trial which aims to assess the clinical- and cost-effectiveness of an intervention designed to enable recovery of outdoor mobility among older adults after hip fracture (the OUTDOOR intervention).

### Methods

This is a protocol for a multi-centre pragmatic parallel group (allocation ratio 1:1) randomised controlled assessor-blinded feasibility trial. Adults aged 60 years or more, admitted to hospital from- and planned discharge to- home, with self-reported outdoor mobility in the three-months pre-fracture, surgically treated for hip fracture, and who are able to consent and participate, are eligible. Individuals who require two or more people to support mobility on discharge will be excluded. Screening and consent (or consent to contact) will take place in hospital. Baseline assessment and randomisation will follow discharge from hospital. Participants will then receive usual care (delivered by physiotherapy, occupational therapy, or therapy assistants), or usual care plus the OUTDOOR intervention. The OUTDOOR intervention includes a goal-orientated outdoor mobility programme (supported by up to six in-person visits), therapist-led motivational dialogue (supported by up to four telephone calls),

**Data Availability Statement:** No datasets were generated or analysed during the current study.

**Funding:** This paper presents independent research funded by the National Institute for Health

Research (NIHR) under its Research for Patient Benefit (RfPB) Programme (Grant Reference Number NIHR204040) and The Royal Osteoporosis Society (Grant Reference Number 518) awarded to KS, EG, CS, SL, SC, CH, FCM, and TS. This work acknowledges the support of the National Institute for Health Research Barts Biomedical Research Centre (NIHR203330). The views expressed are those of the author(s) and not necessarily those of the NIHR or the Department of Health and Social Care. The funder has no competing interests, has had no substantial influence on the planning of the trial, and they will not influence the conduct or reporting of the trial. The funders had no role in study design, data collection and analysis, decision to publish, or preparation of the manuscript. For the purpose of open access, the author has applied a Creative Commons Attribution (CC BY) licence to any Author Accepted Manuscript version arising from this submission. Sponsors URL: http://www.kcl.ac.uk/index.aspx and https://improvinglivesnw.org.uk/about-us/our-nhs-integrated-care-board-icb/.

**Competing interests:** The authors have received grants from the National Institutes of Health Research (NIHR) and The Royal Osteoporosis Society related to this work. This funding provides salary support for DB, and partial salary support for SC, CH, SL, CS, EG and KS. KS receives funding from UKRI for hip fracture health services research. KS is Chair of the Scientific and Publications Committee at the Falls and Fragility Fracture National Audit programme and the Chair-elect of the Scientific Committee of the Fragility Fracture Network. CS, TS, SC receive funding from the National Institutes of Health Research for research not related to the current study. SC is partially supported by the National Institute for Health Research Applied Research Collaboration South West Peninsula. DS, SG, SP, FM declare no additional conflicts of interest. This does not alter our adherence to PLOS ONE policies on sharing data and materials.

supported by a past-patient led video where recovery experiences are shared, and support to transition to independent ongoing recovery. Therapists delivering the OUTDOOR intervention (distinct from those supporting usual care) will receive training in motivational interviewing and behaviour change techniques. Baseline demographics will be collected. Patient reported outcome measures including health related quality of life, activities of daily living, pain, community mobility, falls related self-efficacy, resource use, readmissions, and mortality will be collected at baseline, 6-weeks, 12-weeks, and 6-months (for those enrolled early in the trial) post-randomisation. Exercise adherence (6- and 12- weeks) and intervention acceptability (12-weeks) will be collected. A subset of 20 participants will also support accelerometry data collection for 10 days at each time point.

## Dissemination

The trial findings will be disseminated to patients and the public, health professionals and researchers through publications, presentations and social media channels.

## Trial registration

The trial has been registered at ISRCTN16147125.

## Protocol version

3.0.

## Background and rationale

Around 70,000 older adults experience hip fracture in the United Kingdom (UK) each year [1]. All but those deemed to be at end of life receive surgery, after which there is a 5- to 8-fold increase in all-cause mortality risk by three months after hip fracture [2]. Further, there are reported high rates of transition from independent living to nursing homes among persons with hip fracture [3]. These poor outcomes led 81 global societies to endorse a call to action for ongoing post-acute care of people whose ability to function is impaired following hip fractures [4].

A 2022 review synthesised the evidence from 14 studies which explored 279 patient perspectives of recovery after hip fracture [5]. Across studies, patients considered recovery as a return to pre-fracture activities often requiring outdoor mobility e.g., gardening, shopping, participating in social events such as meeting friends, attending the theatre, and volunteering [5]. These priorities reflect the World Health Organization's definition of functional ability as 'all the health-related attributes that enable people to be and to do what they have reason to value' [6].

An analysis of 99,667 patients with hip fracture in England, Wales and Northern Ireland reported 74% of patients had outdoor mobility pre-fracture, but only 9% of these individuals recovered this mobility by 30 days post-fracture [7]. This proportion increased to 26% by 120 days [8]. Despite this common failure to support 74% of patients with hip fracture to regain their pre-fracture outdoor mobility, research and clinical guidelines focus on acute rehabilitation and short-term outcomes, the focus of community rehabilitation being limited to in-home mobility [9–11]. In the absence of evidence, interventions to improve outdoor mobility are not included as standard in UK community rehabilitation after hip fracture [12]. The

James Lind Alliance identified this absence of evidence as a top 10 priority for lower limb fragility fracture among older adults [13].

Results from two systematic reviews suggested a potential benefit of an outdoor mobility programme which includes: 1) walking, use of assistive devices and transport; and 2) a behaviour change component, on outdoor mobility, physical activity and endurance [14, 15]. The systematic reviews also suggested a potential benefit of these interventions with respect to falls-related self-efficacy (confidence intervals crossed null values for both reviews) [14, 15]. No trial identified in either review included an intervention component targeting concerns about falls. This is despite the reported negative association between concerns about falls and outdoor mobility behaviour [16, 17]. Given the high proportion of patients who do not regain outdoor mobility after hip fracture, rehabilitation explicitly targeting outdoor mobility is needed to enable these older adults to recover activities which they value most.

## Aims and objectives

The overarching aim of this study is to determine the feasibility of a randomised controlled trial (RCT) which aims to assess the clinical- and cost-effectiveness of an intervention designed to enable recovery of outdoor mobility among adults after hip fracture (the OUTDOOR intervention).

The primary objective of this feasibility trial is to determine whether the OUTDOOR intervention is delivered as intended considering the five domains of intervention fidelity (design, training, delivery, receipt, and enactment). The secondary objectives include the assessment of the:

- acceptability of the OUTDOOR intervention to participants and therapists

- barriers and enablers to OUTDOOR intervention delivery

- count of eligible, recruited and retained participants

- acceptability, completeness and descriptive comparison of outcome (including economic and accelerometery) data collection

- count of inadvertent unblinding of outcome assessors

- count of adverse and serious adverse events

- indicative sample size for a definitive trial

## Methods

The protocol is reported in accordance with the Standard Protocol Items: Recommendations for Interventional Trials (SPIRIT) Checklist [18] (S1–S3 Files) and registered at ISRCTN16147125. The trial has received approval from East of England–Essex Research Ethics Committee (REF: 23/EE/0246) and the Health Research Authority. The schedule of events is shown in Fig 1.

### Public and patient involvement

Patient and public involvement (PPI) was conducted from proposal conception onwards. The PPI group 'TROOP': Trauma Rehabilitation (Orthopaedic) research for Older People contributed to trial design during a series of focus groups (outcome measure selection, participant eligibility), the intervention development workshops, and the development of patient facing materials (including a video) for the trial. The active collaboration will continue during the

| Event | Time point for collection | | | | | | |
|---|---|---|---|---|---|---|---|
| | Recruitment | Baseline | Randomisation | Intervention (up to 6 visits over 12 weeks) | 6-weeks | 12-weeks | 6-months* |
| Screening log | X | | | | | | |
| Approach log | X | | | | | | |
| Contact details | X | | | | | | |
| Consent log | X | | | | | | |
| Age | | X | | | | | |
| Sex | | X | | | | | |
| Ethnicity | | X | | | | | |
| Fracture type | | X | | | | | |
| Surgery type | | X | | | | | |
| Abbreviated Mental Test | | X | | | | | |
| Clinical Frailty Scale | | X | | | | | |
| Residence | | X (prefracture) | | | | | |
| Living status | | X (prefracture) | | | | | |
| Mobility | | X (prefracture) | | | | | |
| Discharge direct to home | | X | | | | | |
| EuroQoL EQ-5D-5L | | X | | | X | X | X |
| Nottingham Extended Activities of Daily Living | | X | | | X | X | X |
| Falls Efficacy Scale-International | | X | | | X | X | X |
| Numeric Rating Scale | | X | | | X | X | X |
| University of Alabama Life Space Assessment | | X | | | X | X | X |
| Bespoke resource use form | | X | | | X | X | X |
| Randomisation log | | | X | | | | |
| Treatment -control | | | | X | | | |
| Treatment -intervention | | | | X | | | |
| Treatment logs | | | | X | | | |
| Deviation log | | | | X | | | |
| Treatment audio recordings | | | | X | | | |
| Patient diary | | | | X | | | |
| Length of stay | | X | | | | | |
| Mortality | | | | X | X | X | X |
| Readmission | | | | X | X | X | X |
| Readmission diagnosis (as applicable) | | | | X | X | X | X |
| Completion logs | | | | | | X | |
| Patient semi-structured interviews | | | | | | X | |
| Patient acceptability | | | | | | X | |
| Family/friends engaged | | | | | | X | |
| Therapist semi-structured interviews | | | | | | X | |
| Therapist acceptability | | | | | | X | |

* if the timing of randomisation permits this follow-up within the trial data collection window; orange fill indicates events applicable only to the intervention arm.

**Fig 1. Study time/event matrix.**

feasibility trial, with PB taking a leadership role for TROOP during Trial Management Group (TMG) meetings. TROOP will also meet at regular intervals remotely for 90-minutes to discuss progress, and development and dissemination of plain English summaries of the project findings. Two PPI members independent of TROOP were recruited for the joint Trial Oversight Committee through NIHR People in Research. The UK Standards for Public Involvement will continue to be followed to ensure this collaboration follows best practice [19].

## Design

A multi-centre pragmatic (real world) parallel group (participants allocated equally to one of two groups—ratio 1:1) randomised controlled assessor-blinded feasibility trial (Fig 2).

 * by telephone or MS TEAMS. †if the timing of randomisation permits this follow-up within the trial data collection window. ‡ for 20 participants enrolled for accelerometery data collection circulated by post, and returned by pre-paid courier.

## Setting

Screening and consent processes (or consent to contact) will take place in hospital. For those who provide consent to contact, subsequent consent will be sought once they return home. Physiotherapists/occupational therapists/therapy assistants (hereafter referred to as 'therapists') working outside of the hospitals will support delivery of the OUTDOOR intervention and usual care from a participant's home. Therapists delivering the OUTDOOR intervention will receive training and be distinct from therapists delivering usual care to avoid contamination. We selected sites across the UK to ensure feasibility is assessed in a diverse range of contexts and local populations.

## Eligibility criteria

### Patient participant inclusion.

- Age 60 years or more

- Admitted to hospital from (and planned discharge to) home

- Self-reported outdoor mobility in the three-months pre-fracture

- Surgically treated for hip fracture

- Able to consent and participate in the trial

### Patient participant exclusion.

- Age less than 60 years

- Admitted to hospital from (and/or planned discharge to) a care home

- No self-reported outdoor mobility in the three-months pre-fracture

- Not surgically treated for hip fracture

- Require two or more persons to support mobility on discharge

- Not able to consent and participate in the trial

**Justification for patient participant eligibility criteria.** Sixty-years of age was selected to align with the National Hip Fracture Databases definition of the target population of older

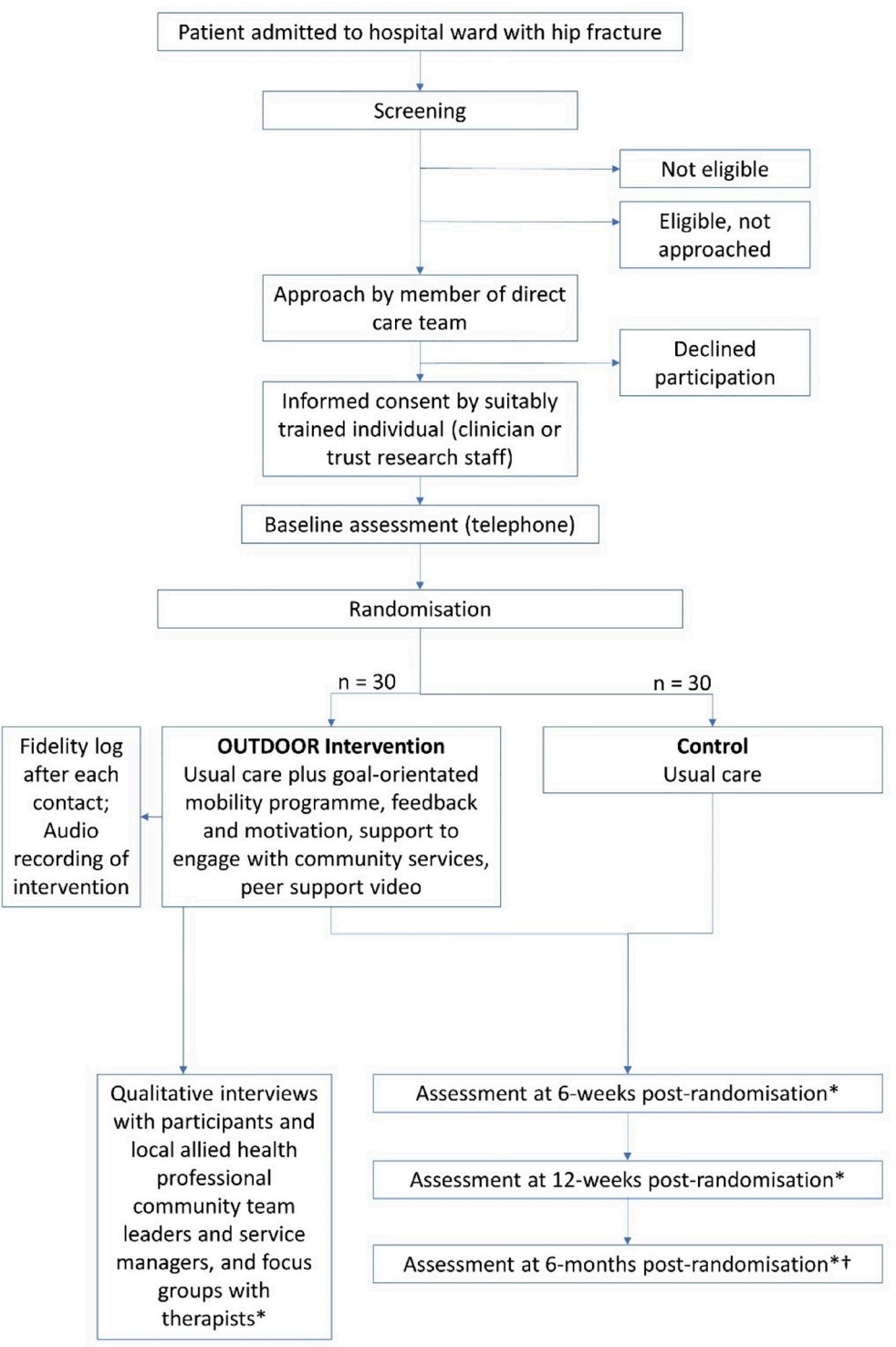

**Fig 2. Flow diagram for the OUTDOOR feasibility RCT.**

people who incur fragile hip fracture (average age at hip fracture is 83 years for women and 84 years for men [11, 20]. The requirements of home residency, prior outdoor mobility, and ability to consent and participate were selected to ensure the safety of participants (as the intervention profile would vary for those admitted from care home, without outdoor mobility and/or capacity to consent and participate). Surgical management was a requirement for participation as non-surgical management is reserved for around 2% of hip fracture patients in the UK who are often at the end of life. Finally, the exclusion of potential participants who require two or more persons to support mobility on discharge was selected as the intervention would require visits by two therapists to support implementation which on consultation with community therapists would not be feasible.

**Professional eligibility.** Therapists involved in the OUTDOOR intervention arm of the feasibility trial will be invited to a focus group at the end of the intervention delivery period. Managers who oversee services involved in the delivery of the OUTDOOR intervention will be invited to an interview.

## Recruitment

A member of the site clinical team will screen potential participants for eligibility during the inpatient stay after hip fracture surgery. A suitably trained individual will provide information leaflets and answer questions and obtain written consent (or not) from eligible participants (or consent to contact on discharge home). For participants who provide consent to contact on discharge home, they will be followed up by the local site community teams to answer any questions and obtain written consent (or not) to participate. Age, sex and self-reported ethnicity will be collected for patients screened but who do not subsequently provide consent, allowing generalisability of the randomised participants as well as the screened eligible patients to be interpreted. Reasons for ineligibility will be documented together with reasons volunteered by potential participants for declining to do so. Participants recruited whose circumstances change (e.g., planned discharge to home but then discharged to nursing/residential care) will be withdrawn before randomisation. They will not be replaced. Participation in accelerometery data collection will be optional for up to 20 participants (10 intervention group, 10 control group). Reasons for declining to participate in accelerometery data collection will be documented if offered.

Therapists involved in delivery of the OUTDOOR intervention will be invited to take part in semi-structured focus groups scheduled after the end of the intervention stage on treatment acceptability and fidelity. They will be provided with a participation information leaflet during therapist training prior to the start of the trial and asked to provide consent to contact. Towards the end of the study, a member of the research team will follow-up to answer any questions prior to obtaining written informed consent. Managers of services involved in the delivery of the OUTDOOR intervention will be invited to participate in semi-structured interviews on treatment acceptability and fidelity. A member of the research team will provide a participation information leaflet via email after which a member of the research team will follow up to answer any questions prior to seeking written informed consent.

## Randomisation and allocation procedure

Participants will be allocated in a 1:1 ratio to OUTDOOR intervention or control (usual care) groups. The Exeter Clinical Trials Unit (ExeCTU) will generate the allocation sequence, stratified by recruiting site, using minimisation (with random element) for pre-fracture mobility (freely mobile outdoors without aids, mobile outdoors with one aid, mobility outdoors with two aids or frame). Randomisation of a participant will be completed by a member of the

research team after baseline data collection is completed following discharge from hospital, using a secure internet-based system, developed and maintained by ExeCTU to ensure allocation concealment. Treatment allocation will be linked to a participant identification number and the clinical team will be notified.

## Intervention

**Theoretical frameworks.**  Webber's theoretical framework for mobility in older adults was employed [21]. This framework defines mobility as the ability to move oneself (by walking, with assistive devices or transport) within community environments that expand from a room within a person's home to outdoors–a person's garden or driveway, to their local neighborhood of nearby streets and parks, to the service community (banks, shops, GP office), and then the surrounding area and beyond [21]. The concept of mobility is portrayed through five fundamental determinants–physical, cognitive, psychosocial, environmental, and financial [21]. These determinants will hold different degrees of importance depending on the person, where a person is going and how, but the determinants are related to each other and become more complex the further a person travels from their own home [21]. This framework enables building on the previous evidence by considering mobility in the current proposed trial as walking, use of assistive devices and transport.

Operationalising Webber's determinants required consideration of two additional theories. First, the COM-B model of behaviour change targeting participants need for capability, opportunity, and motivation to generate and maintain a desired behaviour [22]. Second, Normalisation Process Theory which looks at how to embed a practice into 'work as usual' through four components [23]. Coherence relates to understanding and making sense of a practice (here 'outdoor mobility'), cognitive participation–engaging and participating with the practice, collective action–the joint 'work' needed to enact the practice, and reflexive monitoring–reflecting and appraising the practice over time to ensure it becomes routinely embedded [23]. This theory was considered essential for designing an intervention with future sustainability in mind. Each of these three theories were incorporated into the development of programme theories and logic model for the proposed OUTDOOR intervention.

**Development.**  Two remote intervention development workshops were convened comprised of patient and carer representatives, physiotherapists, triallists, implementation and behaviour change scientists, as well as a panel of experts in rehabilitation for patients after hip fracture from the UK, Denmark, Norway, Spain, USA, Canada, Brazil and Australia (n = 20 participants). At the workshops, the rationale and key question (*what intervention would enable older adults to take part in activities outside the home after hip fracture*?), data/results from existing evidence, including results of two systematic reviews, outline of current UK provision, proposed theoretical frameworks, draft programme theories and logic model were presented. A nominal group technique was then followed inviting participants to generate intervention ideas silently prior to sharing in a round-robin format [24]. All ideas were documented. Participants sought and provided clarification on shared ideas. Top ideas were then prioritised for incorporation into the subsequent draft programme theories and logic model which were finalised after the second workshop (Fig 3) [24].

**Programme theories.**

1. For older adults who were mobile outdoors prior to hip fracture, health-related quality of life (HRQoL) post-fracture is enhanced by regaining outdoor mobility;

2. Achievement of physical, psychosocial, and cognitive capability to go outdoors requires tailored, structured and graded support with mechanisms to monitor and preserve gains.

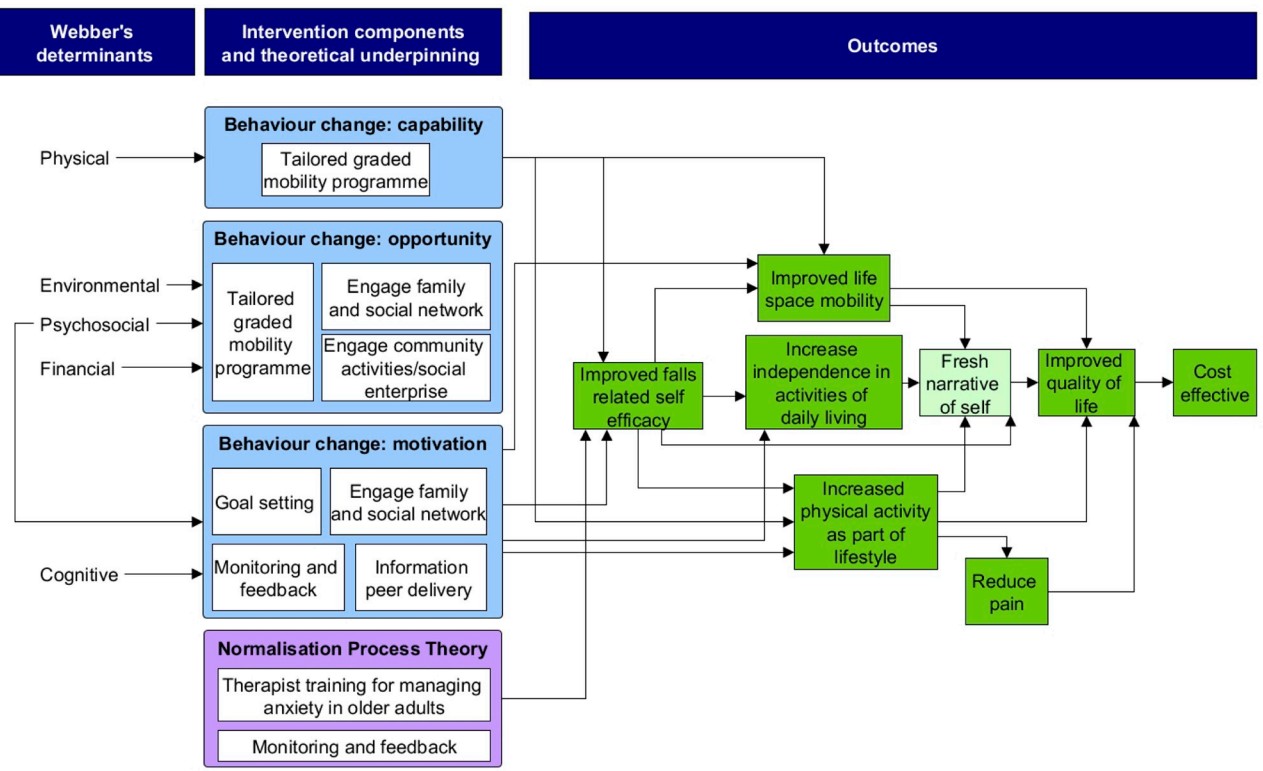

**Fig 3. Logic model for the OUTDOOR intervention.**

**Intervention description.** The intervention will start approximately 30 days after a participant returns home and end when they have received a maximum of 6 visits within a maximum of 12 weeks, unless their mobility end goal is achieved before then. From 30-days to 120-days (12-weeks later) was selected given the potential to increase the proportion of patients recovering outdoor mobility in this time frame (with usual care an increase reported from 9% at 30-days to 26% at 120-days [7, 8]. It will be delivered by therapists. Online intervention training (2 hours duration) will be provided to therapists prior to delivery.

Intervention participants will receive usual care and:

1. *Motivation—social support*

   Participants will be provided on-going access (via therapist-held tablet and website link) to a video of older adults who incurred hip fracture, sharing their experience of recovery. The video includes discussion by four older adults about their *capability*, *opportunity*, and *motivation* to generate and maintain their desired behaviour of outdoor mobility (including walking, taking public transport) after hip fracture. Use of this video targets motivation through specific behaviour change techniques of *opportunity for social comparison* and *social support (practical, general, and emotional)* [22].

2. *Goal-orientated mobility programme*

   Participants will have a telephone call/Microsoft TEAMS call with a therapist to set a patient-determined outdoor mobility programme goal, which is both meaningful to the participant and deemed achievable within the scope of the programme by the therapist. The 'programme goal' will be broken down into a maximum of four 'intermediate goals' by the

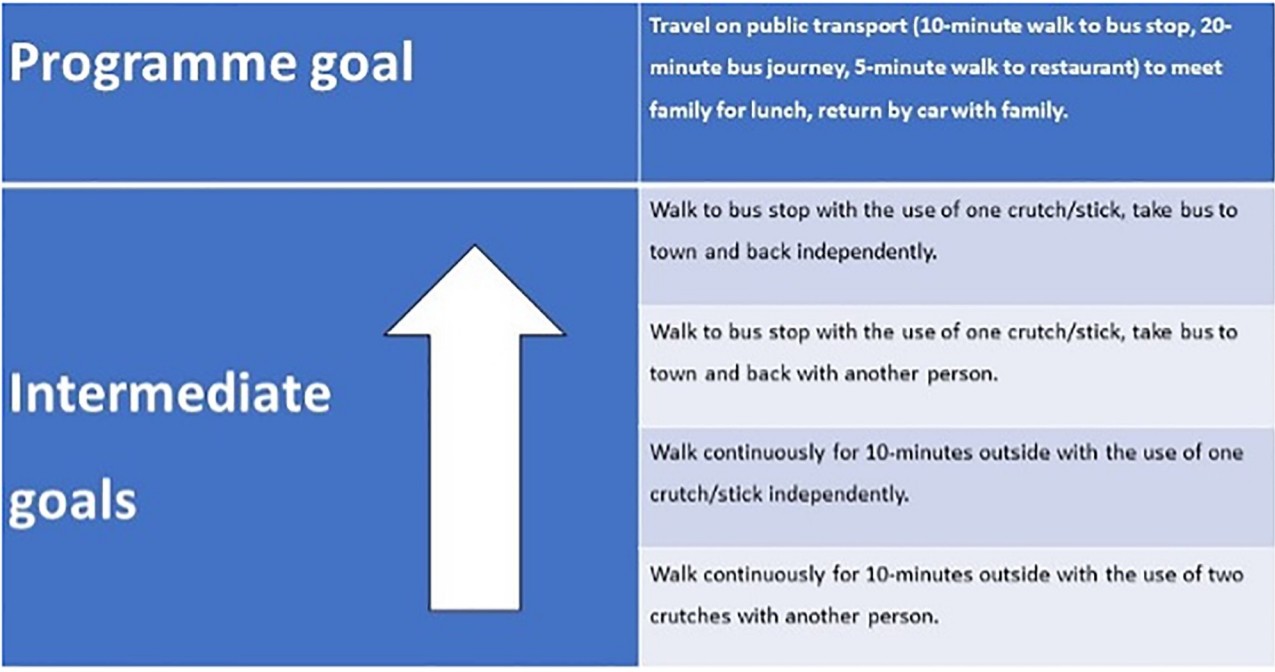

**Fig 4. Example programme and intermediate goals.**

therapist. These intermediate goals will promote movement through life spaces increasingly further from a participant's home and towards their programme goal. These goals will be individually tailored to account for a participants pre-fracture abilities. An example of a programme goal and related intermediate goals is shown in Fig 4.

The programme will be supported by a maximum of six therapist home visits. This will enable a treating therapist to provide supervised support for practice and progression of intermediate goals. Therapists will employ motivational interviewing techniques (engage, focus, evoke and plan) during supervised sessions [25]. The participant will be asked to practice independently and/or with family/friends between therapist visits. They will be asked to document this practice in a diary (diary specifies the goal, whether completed or not, independently or with another person, and how they felt during and after). The goal-orientated mobility programme targets behaviour change techniques related to capability (*commitment*, *behavioural contract*), opportunity (*action planning*, *habit formation*), and motivation (*goal setting*) for outdoor mobility at increasing distances from home [22].

3. ***Motivation—additional professional support***
   In between therapist home visits, the participant will be supported by up to four telephone/ Microsoft TEAMS calls. These calls will be structured around the participant diary and look to reinforce motivational interviewing strategies employed during supervised sessions. This additional support encourages motivation with behaviour change techniques such as *feedback on behaviour*, *others monitoring with awareness*, and *self-monitoring* of both the behaviour and outcome of the behaviour [22].

4. ***Transition to independence***
   During home visits and/or telephone calls, therapists will plan for ongoing recovery after the goal-orientated mobility programme ends. This planning will include supporting

participants to engage with their local community activities and social enterprise groups. The availability of activities and groups will vary according to the participant's place of residence. The activities and groups of interest will be determined by the participant depending on their personal preferences, and participation should be encouraged by the therapist. This transition to independence is supported by behaviour change techniques such as *social support*, *action planning*, *verbal persuasion to boost self-efficacy* and *restructuring of the social environment* [22].

5. ***Additional therapist training***
   As part of the intervention, therapists will receive some training in motivational interviewing [25] and behaviour change techniques [22]. Motivational interviewing will equip therapists with the strategies to reinforce engagement, identify the change required to achieve goals, evoke motivation for change by e.g., shifting to a greater stage of readiness ("I wish" to "I will"), and plan considering where, when how and with whom.

   In particular, the training will focus on strategies for supporting older adults with concerns about falls related to outdoor mobility. As part of intervention development, a systematic review of factors prognostic of concerns about falls following hip fracture was completed [26]. Factors amenable to change were identified for therapists to target through motivational interviewing. These factors included fatigue, safe mobility, consequences of not moving, encouragement and feedback, locus of control, self-confidence, and worries about future and past. The training will support therapists to align these factors to the COM-B model of behaviour change and appropriate behaviour change techniques that can be employed together with motivational interviewing [22]. Therapists will observe motivational interviewing in practice (via a series of videos recorded with actors playing the role of an older person who experienced hip fracture), practice motivational interviewing themselves (with case studies provided and expert feedback), and be provided with access to a recording of the training, example videos, and cases after the session. They will be offered a subsequent revision training session that is therapist-led to discuss any concerns they have in putting motivational interviewing into practice.

## Usual care

The intervention and comparator group will both receive usual care. A survey of UK community rehabilitation for older adults after hip fracture identified two models for those discharged home: early supported discharge with community rehabilitation for four to six weeks or, discharged home under GP care with referral to community services as needed [27]. Where provided, physiotherapy focused on strengthening exercises (100%), progressive resistance training (91%), weight-bearing exercises (95%), gait training (79%), exercise sheets (91%), encouragement of walking and climbing stairs (98%), and transferring (91%) (mode 30 minutes weekly/fortnightly for four to six weeks). Occupational therapy focused on transfer assessment, activities of daily living (e.g., grooming/personal hygiene, dressing, toileting/continence, transferring, and eating), home environment and social support (mode 60 minutes weekly/ fortnightly for four to six weeks). Outdoor mobility and behaviour change techniques [22] were not included in routine community rehabilitation.

## Data collection and outcomes

Data will be collected at baseline, six weeks post-randomisation, 12-weeks post-randomisation, and six-months post-randomisation (if the timing of randomisation permits this follow-up within the trial data collection window).

The following participant characteristics will be collected: age, sex, self-reported ethnicity, Abbreviated Mental Test [28], Clinical Frailty Scale [29], living status (lives alone, with independent spouse, with dependent spouse, with family, with other), accommodation type (house/ground floor apartment with step free access, house/ground floor apartment with step access, 1st floor or higher apartment with lift access, 1st floor or higher with no lift access, sheltered accommodation), pre-fracture mobility (freely mobile outdoors without aids, mobile outdoors with one aid, mobility outdoors with two aids or frame), fracture type, surgery type, length of stay (from admission to discharge from acute hospital), and any additional inpatient stay following discharge from acute hospital.

The acceptability, completeness, and descriptive comparison of patient reported outcome data collection will be assessed through collection of patient-reported outcome measures (PROMs) which satisfy the core outcome set for hip fracture trials namely HRQoL (EuroQoL EQ-5D-5L [30]), activities of daily living (Nottingham Extended Activities of Daily Living [31]), pain (Numeric Rating Scale [32]), community mobility (University of Alabama Life Space Assessment [33]), and mortality. Additional quantitative outcomes captured will include Falls related self-efficacy (Falls Efficacy Scale-International [34]); hospital readmissions and reason and resource use using a bespoke data collection form. The assessor facilitating collection of PROMs will be blind to group allocation.

For participants who do not speak English, PROMs which have an established translated, validated, and (where applicable) culturally adapted version in the appropriate language will be circulated by post to the participant with a pre-paid envelope for return. Participant characteristics and additional quantitative outcomes will be collected using local interpreting services.

Research grade wrist-worn triaxial accelerometers (watch without a face) (GENEActiv www.activinsights.com) will be circulated by post for 10-days of wear at baseline, and at 6-weeks, 12-weeks, and 6-months post-randomisation for the first 20 participants (10 intervention group, 10 control group) who consent to accelerometery data collection. Ten days wear is recommended to enable reliable estimation of habitual physical activity, especially between day variability [35].

To assess fidelity of delivery of the intervention, we will audio record intervention sessions between each participant and their supporting therapist. The audio recording will enable assessment of the five intervention fidelity domains (design, training, delivery, receipt, and enactment) identified by the National Institute of Health Behaviour Change Consortium [36]. Fidelity assessment will be supplemented by study-specific questionnaires completed by the therapists after each supervised session for OUTDOOR intervention and usual care groups (to monitor for contamination between OUTDOOR intervention and comparator arms). For those in the OUTDOOR intervention arm, we will capture whether participants engaged family/friends (yes/no).

Exercise adherence (Exercise Adherence Rating Scale) for the OUTDOOR intervention arm will be assessed at six- and 12- weeks [37]. Participant and therapist acceptability will be assessed with the use of the Theoretical Framework of Acceptability questionnaire at the end of the trial [38, 39].

Qualitative data on perceived barriers and facilitators to OUTDOOR intervention delivery will be captured through purposively sampled (site, pre-fracture mobility, living status, accommodation type, sampled by the research team) remote (telephone/MS TEAMS) semi-structured interviews between a member of the research team and participants. Data collection will continue until no new themes are identified or 50% of participants have been interviewed [40]. Therapists will contribute their perspectives on barriers and facilitators to intervention delivery in one of four 60-minute online focus groups via Microsoft TEAMS facilitated by the

research team. We will target recruitment of at least 80% of the therapists involved in intervention delivery at each site in each focus group. We will also extend invitations for interviews between a member of the research team and people at higher organisational levels (e.g., local allied health professional community team leaders and service managers) for their perspectives on potential barriers and facilitators. Topic guides for interviews and focus groups are available in S4 File.

## Adverse events

Participant safety will be determined through the reporting of adverse events (AE) and serious adverse events (SAE). AEs to be collected and reported will include an exacerbation of a pre-existing illness, a fall that does not require hospitalisation, an increase in the frequency or intensity of a pre-existing episodic event or condition, and/or continuous persistent disease/ symptom present at baseline that worsens. A SAE is an untoward occurrence that results in death, is life threatening at the time of the event, requires unplanned hospitalisation or prolongation of an existing hospitalisation, results in persistent or significant disability or incapacity, and/or a personal data breach. Other 'important medical events' may be considered serious if they jeopardize the participant or require an intervention to prevent one of the aforementioned SAEs. Where any AE/SAE occur, the team will adhere to guidelines for the reporting to the medical and research team, who will subsequently assess relatedness to the intervention and report to the relevant sponsors and regulators. The period for AE/SAE reporting will be from randomisation until final follow-up (6-months if the timing of randomisation permits 6-month follow-up within the trial data collection window, 12-weeks otherwise).

## Sample size

A recruitment target of 60 participants (30 per treatment arm) will allow an overall retention rate at 12 weeks to be estimated with precision of ±11%, using an exact 95% confidence interval, based on previously observed retention rates of ~80% for the same population in the community setting [41]. Assuming a non-differential retention rate of 80% at 12-week follow-up, this target will provide follow-up outcome data on ~24 participants per group.

For therapists and managers, the target recruitment is 80% from each site. This is estimated to be 12 therapists and four managers across sites.

## Data analysis

**Quantitative.** A statistical analysis plan will be finalised ahead of database locking and reporting will follow the CONSORT guidance for pilot and feasibility studies [42]. Primary (descriptive only) analyses will follow the principles of intention to treat [43]. A CONSORT flow diagram will display data from screening, recruitment and follow-up logs enabling estimation of eligibility, recruitment, consent and follow-up rates [42]. Confidence intervals for recruitment and retention rates will be produced to inform assumptions for planning the definitive trial. Completion rates will be estimated for outcome measures at each time-point, including resource use and accelerometery data. Participants' baseline characteristics will be summarised by allocated group with descriptive statistics (measures of central tendency and dispersion). Patient-reported outcomes, acceptability, and accelerometery (days of valid wear, frequency and volume of activity, temporal distribution of activity, estimate of frequency of active outdoor events) measures will be summarised by allocated group at each follow-up, with descriptive statistics (measures of central tendency and dispersion). Between-group differences, including changes from baseline, will be reported for the PROMs and accelerometery

with corresponding confidence intervals. For participants who do not speak English, the count of pseudo-anonymised PROMs circulated and returned by post will be documented. The content of the PROMs will be described narratively ensuring participant anonymity is preserved (the language versions circulated will not be specified in reporting). The analyses will be completed at the end of the trial (no planned interim analyses) by a trial statistician who will be blind to group allocation at least until the statistical analysis plan is drafted, using well-validated statistical packages.

**Qualitative.** Qualitative data transcribed verbatim from semi-structured interviews and focus groups will be analysed using a thematic analysis approach [44]. A random sample of 10% intervention audio recordings will be sampled to assess fidelity. High fidelity will be considered as ≥80% of intervention core components fully delivered by the therapist within each session. Qualitative analysis will be completed by a member of the research team as data becomes available (completion of interviews/focus groups and/or availability of audio recordings to assess fidelity).

## Progression criteria

To mitigate the risk of ongoing uncertainty at the end of the feasibility trial, we propose progression criteria outlined in Table 1 [45]. If the accelerometery criteria suggest STOP at the end of the feasibility RCT but the remaining criteria suggest AMEND/GO, then a definitive trial will be pursued with accelerometery excluded.

**Table 1. Progression criteria.**

|  | GO | AMEND | STOP |
|---|---|---|---|
| Recruitment | ≥40% eligible | 21–39% eligible | ≤20% eligible |
| Recruitment | ≥50% eligible recruited | 31–49% eligible recruited | ≤30% eligible recruited |
| Randomisation | ≥70% of those recruited are randomised | 49–70% of those recruited are randomised | ≤48% of those recruited are randomised |
| Acceptability*, participants | Median of ≥28 | Median of 24–27 | Median of <24 |
| Acceptability*, therapists | Median of ≥28 | Median of 24–27 | Median of <24 |
| Fidelity | ≥80% sessions included all intervention components as described | 51–79% sessions included all intervention components as described | ≤50% sessions included all intervention components as described |
| Outcome, 12-weeks | ≥80% completeness of EQ5D at 12-week follow-up | 51–79% completeness of EQ5D at 12-week follow-up | ≤50% completeness of EQ5D at 12-week follow-up |
| Outcome, 6-months† | ≥70% completeness of EQ5D at 6-month follow-up | 31–69% completeness of EQ5D at 6-month follow-up | ≤30% completeness of EQ5D at 6-month follow-up |
| Accelerometery recruitment | ≥50% recruited to accelerometry. | 31–49% recruited to accelerometry. | ≤30% recruited to accelerometry. |
| Accelerometery Acceptability | ≥50% enrolled considered accelerometry acceptable. | 31–49% enrolled considered accelerometry acceptable. | ≤30% enrolled considered accelerometry acceptable. |
| Accelerometery, 12-weeks | ≥80% accelerometers returned with data for 5 or more days at 12-weeks. | 51–79% accelerometers returned with data for 5 or more days at 12-weeks. | ≤50% accelerometers returned with data for 5 or more days at 12-weeks. |
| Accelerometery, 6-months | ≥70% accelerometers returned with data for 5 or more days at 6-months. | 31–69% accelerometers returned with data for 5 or more days at 6-months. | ≤30% accelerometers returned with data for 5 or more days at 6-months. |

*Theoretical Framework of Acceptability questionnaire is comprised of eight questions (affective attitude, burden, perceived effectiveness, intervention coherence, self-efficacy, opportunity costs, general acceptability) each a five-point scale (score range 8–40).

†if the timing of randomisation permits this follow-up within the trial data collection window.

## Monitoring

The trial will be conducted in compliance with the approved protocol, to Good Clinical Practice, the UK General Data Protection Regulation and Data Protection Act (2018), the local Information Governance Policy, the UK Policy Framework for Health and Social Care Research, the sponsor's Standard Operating Procedures, the Mental Capacity Act 2005, and other applicable regulations as required. All monitoring for this trial will be done remotely and centrally at the ExeCTU throughout the conduct of the trial, with onsite monitoring occurring for cause-triggered visits only. A joint Trial Oversight Committee comprised of members of the research team and independent members (chair, statistician, expert members, public representatives) will provide advice, data monitoring (screening and recruitment rates, accruing outcome data), quality assurance, and safety monitoring (number, nature and outcomes for all serious adverse events). The committee may include open and closed sessions. Closed sessions will not be attended by blinded members of the research team and may be used for data monitoring and/or other discussions at the discretion of the chair. The committee will be asked to recommend any necessary actions. It is anticipated that the committee will meet at least biannually during the trial period. The charter is available from the lead author on request.

## Dissemination

The results of the study will be summarised in plain English and made available on the TROOP PPI group webpage (www.ppitroop.co.uk) and Twitter page (@TROOP_PPI) as well as via charity newsletters. Participants will be offered the option of having the plain English summary posted directly to them during the informed consent process. The results of the study will be published in an open-access peer reviewed journal. The findings will be presented at national and international conferences. We will submit findings to guideline committees. We will share results directly to UK clinicians via the Chartered Society of Physiotherapy, Royal College of Occupational Therapists, British Geriatrics Society and Fragility Fracture Network UK. Following publication of the primary paper, anonymised electronic data will be exported and stored alongside anonymised transcriptions of interviews on the King's Open Research Data System (https://www.kcl.ac.uk/researchsupport/managing/preserve), with proof of ethical approval as a condition of access.

## Supporting information

**S1 File. SPIRIT checklist.**
(PDF)

**S2 File. Protocol.**
(PDF)

**S3 File. Consent form.**
(PDF)

**S4 File. Topic guides.**
(PDF)

## Acknowledgments

We are grateful to the input from patient and public members of the involvement group TROOP (https://www.ppitroop.co.uk/) for their support in the design of this trial, to the participants in the qualitative interview studies and intervention development workshops which

underpin the intervention, and to the volunteers who supported the development of the peer video. We acknowledge the study Principal Investigators are Nicola Sinclair, Stephanie Tuck, Zita Lodge, Rebecca Wood, and Jonathan Evans. Sandra Houston, Amina Mourad, and Rhian Milton-Cole will serve as Associate Principal Investigators.

## Author Contributions

**Conceptualization:** Katie J. Sheehan, Denise Bastas, Stefanny Guerra, Siobhan Creanor, Claire Hulme, Sallie Lamb, Finbarr C. Martin, Catherine Sackley, Toby Smith, Philip Bell, Melvyn Hillsdon, Sarah Pope, Heather Cook, Emma Godfrey.

**Funding acquisition:** Katie J. Sheehan, Siobhan Creanor, Claire Hulme, Sallie Lamb, Finbarr C. Martin, Catherine Sackley, Toby Smith, Emma Godfrey.

**Methodology:** Katie J. Sheehan, Denise Bastas, Stefanny Guerra, Siobhan Creanor, Claire Hulme, Sallie Lamb, Finbarr C. Martin, Catherine Sackley, Toby Smith, Philip Bell, Melvyn Hillsdon, Sarah Pope, Heather Cook, Emma Godfrey.

**Writing – original draft:** Katie J. Sheehan, Denise Bastas, Stefanny Guerra.

**Writing – review & editing:** Katie J. Sheehan, Denise Bastas, Stefanny Guerra, Siobhan Creanor, Claire Hulme, Sallie Lamb, Finbarr C. Martin, Catherine Sackley, Toby Smith, Philip Bell, Melvyn Hillsdon, Sarah Pope, Heather Cook, Emma Godfrey.

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
