## [Decision Letter · Decision Letter 0]

28 May 2024

PONE-D-24-06214Protocol for a Feasibility Randomised Controlled Trial of the ‘Outdoor’ Mobility Intervention for Older Adults after Hip FracturePLOS ONE

Dear Dr. Sheehan,

Thank you for submitting your manuscript to PLOS ONE. After careful consideration, we feel that it has merit but does not fully meet PLOS ONE’s publication criteria as it currently stands. Therefore, we invite you to submit a revised version of the manuscript that addresses the points raised during the review process. As indicated by the second reviewer, it is critical that 'National Hip Fracture Databases ' be clarified in context.

We look forward to receiving your revised manuscript.

Kind regards,

Diphale Joyce Mothabeng, PhD

Academic Editor

PLOS ONE

Journal Requirements:

2. PLOS requires an ORCID iD for the corresponding author in Editorial Manager on papers submitted after December 6th, 2016. Please ensure that you have an ORCID iD and that it is validated in Editorial Manager. To do this, go to ‘Update my Information’ (in the upper left-hand corner of the main menu), and click on the Fetch/Validate link next to the ORCID field. This will take you to the ORCID site and allow you to create a new iD or authenticate a pre-existing iD in Editorial Manager. Please see the following video for instructions on linking an ORCID iD to your Editorial Manager account: https://www.youtube.com/watch?v=_xcclfuvtxQ".

[We are grateful to the input from patient and public members of the involvement group TROOP (https://www.ppitroop.co.uk/) for their support in the design of this trial, to the participants in the qualitative interview studies and intervention development workshops which underpin the intervention, and to the volunteers who supported the development of the peer video. We acknowledge the study Principal Investigators are Nicola Sinclair, Stephanie Tuck, Zita Lodge, Rebecca Wood, and Jonathan Evans. Sandra Houston, Amina Mourad, and Rhian Milton-Cole will serve as Associate Principal Investigators. This work acknowledges the support of the National Institute for Health Research Barts Biomedical Research Centre (NIHR203330).]

 [This paper presents independent research funded by the National Institute for Health Research (NIHR) under its Research for Patient Benefit (RfPB) Programme (Grant Reference Number NIHR204040) and The Royal Osteoporosis Society (Grant Reference Number 518) awarded to KS, EG, CS, SL, SC, CH, FCM, and TS. The views expressed are those of the author(s) and not necessarily those of the NIHR or the Department of Health and Social Care. The funder has no competing interests, has had no substantial influence on the planning of the trial, and they will not influence the conduct or reporting of the trial. The funders had no role in study design, data collection and analysis, decision to publish, or preparation of the manuscript. For the purpose of open access, the author has applied a Creative Commons Attribution (CC BY) licence to any Author Accepted Manuscript version arising from this submission. Sponsors URL: http://www.kcl.ac.uk/index.aspx and https://improvinglivesnw.org.uk/about-us/our-nhs-integrated-care-board-icb/.]

[I have read the journal's policy and the authors of this manuscript have the following competing interests: The authors have received grants from the National Institutes of Health Research (NIHR) related to this work. This funding provides salary support for DB, and partial salary support for SC, CH, SL, CS, EG and KS. KS receives funding from UKRI for hip fracture health services research. KS is Chair of the Scientific and Publications Committee at the Falls and Fragility Fracture National Audit programme and the Chair-elect of the Scientific Committee of the Fragility Fracture Network. CS, TS, SC receive funding from the National Institutes of Health Research for research not related to the current study. SC is partially supported by the National Institute for Health Research Applied Research Collaboration South West Peninsula. DS, SG, SP, FM declare no additional conflicts of interest. ]. 

7. Ethics statement appears in the Methods section of the manuscript AND at the end of the manuscript:

Your ethics statement should only appear in the Methods section of your manuscript. If your ethics statement is written in any section besides the Methods, please delete it from any other section.

Reviewers' comments:

Reviewer's Responses to Questions

**Comments to the Author**

1. Does the manuscript provide a valid rationale for the proposed study, with clearly identified and justified research questions?

Reviewer #1: Yes

Reviewer #2: Yes

2. Is the protocol technically sound and planned in a manner that will lead to a meaningful outcome and allow testing the stated hypotheses?

Reviewer #1: Yes

Reviewer #2: Partly

3. Is the methodology feasible and described in sufficient detail to allow the work to be replicable?

Reviewer #1: Yes

Reviewer #2: No

4. Have the authors described where all data underlying the findings will be made available when the study is complete?

Reviewer #1: Yes

Reviewer #2: Yes

5. Is the manuscript presented in an intelligible fashion and written in standard English?

Reviewer #1: Yes

Reviewer #2: No

6. Review Comments to the Author

You may also provide optional suggestions and comments to authors that they might find helpful in planning their study.

Reviewer #1: - Figure 1 attachment is not systematically displayed. Figure 2 is number 1 displayed

- Design: to describe it, explain the pragmatic parallel group allocation.

- I did not see how the assessors will be blinded. Please clarify this section

- 135 - 137: the highlight of the figure is confusing to the reader. I suggest it is added under the figure itself not in text.

- 186: participants changed from home discharge to nursing home will be withdrawn "and not replaced" - will this not reduce the sample size?

- Settings: Description of the study settings is unclear.

- 351 - 354: Check the number of brackets used

- 406: Fullstop before the reference

Reviewer #2: The study/protocol addresses an important issue in senior wellness. I have a few clarifying questions/comments:

1. L78

An analysis of the National Hip Fracture Database (England, Wales and Northern Ireland) reported 74% of patients had outdoor mobility pre-fracture, but only 9% of these individuals recovered this mobility by 30 days post-fracture…

- It would be useful to give a brief explanation of what the National Hip Fracture Database is (and relevance to this study)

- Who are the 9%, 26% that are mobilizing outdoors by 30 days, 120 days? Some description of these profiles is important in building the rationale for this work. E.g., how does age feature in this? Are the 9%, 26% younger with fewer or no comorbidities?

2. L255

The authors state, “trained therapists” - who are/could be the trained therapists who will deliver the intervention? Please specify eg, physios, OTs, PTA, PT/OT Assistants, etc.

3. Line (L) 253 - L257 The rationale for starting the intervention 30 days after return home is not stated.

Also, to clarify, the duration of the intervention will be a maximum of 6 visits within a maximum of 12 weeks unless the end-goal is achieved before then. Is this correct?

If so, please clarify this; it’s a bit confusing.

4. “Sixty-years of age was selected to align with the National Hip Fracture Databases definition of the target population(20).”

What is National Hip Fracture Databases rationale for selecting 60 years as the target population age? How do risk factors for falls change with advancing age? And if 60 years is an important for comparison purposes, how is this likely (or not) to bias the results. These issues have not been adequately addressed in the protocol.

5. There’s inconsistency in the spelling of “accelerometry”

6. Semi-structured interview and focus group guides were not provided.

7. PLOS authors have the option to publish the peer review history of their article (what does this mean?). If published, this will include your full peer review and any attached files.

Reviewer #1: **Yes: **Dr Nontembiso Magida, University of Pretoria

Reviewer #2: No

---

## [Author Response · Author response to Decision Letter 0]

30 May 2024

Reviewer #1: 

1. Figure 1 attachment is not systematically displayed. Figure 2 is number 1 displayed.

a. Updated. 

2. Design: to describe it, explain the pragmatic parallel group allocation.

a. Updated to read: ‘This is a protocol for a multi-centre pragmatic (real world) parallel group (participants allocated equally to one of two groups - ratio 1:1) randomised controlled assessor-blinded feasibility trial.’

3. I did not see how the assessors will be blinded. Please clarify this section.

a. The following statement is included under ‘data collection and outcomes’: ‘The assessor facilitating collection of PROMs will be blind to group allocation.’

4. 135 - 137: the highlight of the figure is confusing to the reader. I suggest it is added under the figure itself not in text.

a. This is journal policy so will leave to editor discretion. 

5. 186: participants changed from home discharge to nursing home will be withdrawn "and not replaced" - will this not reduce the sample size?

a. Yes. We do not anticipate many withdrawals prior to randomisation. We will monitor as part of our objective to assess the count of eligible, recruited, and retained participants. 

6. Settings: Description of the study settings is unclear.

a. We updated the setting to read: ‘Screening and consent processes (or consent to contact) will take place in hospital. For those who provide consent to contact, subsequent consent will be sought once they return home. Physiotherapists/occupational therapists/therapy assistants (hereafter referred to as ‘therapists’) working outside of the hospitals will support delivery of the OUTDOOR intervention and usual care from a participant’s home. Therapists delivering the OUTDOOR intervention will receive training and be distinct from therapists delivering usual care to avoid contamination. We selected sites across the UK to ensure feasibility is assessed in a diverse range of contexts and local populations.’

7. 351 - 354: Check the number of brackets used.

a. Reviewed and confirmed correct. 

8. 406: Fullstop before the reference.

a. Updated. 

Reviewer #2: 

1. The study/protocol addresses an important issue in senior wellness. I have a few clarifying questions/comments. L78: An analysis of the National Hip Fracture Database (England, Wales and Northern Ireland) reported 74% of patients had outdoor mobility pre-fracture, but only 9% of these individuals recovered this mobility by 30 days post-fracture…It would be useful to give a brief explanation of what the National Hip Fracture Database is (and relevance to this study).

a. Thank you for this comment – we updated the background to read ‘An analysis of 99,667 patients with hip fracture in England, Wales and Northern Ireland reported 74% of patients had outdoor mobility pre-fracture, but only 9% of these individuals recovered this mobility by 30 days post-fracture(7). This proportion increased to 26% by 120 days(8).’ The references will take the reader to analyses of the National Hip Fracture Database (NHFD)– a national audit for England, Wales and Northern Ireland with >95% case ascertainment which has been running since 2007. 

2. Who are the 9%, 26% that are mobilizing outdoors by 30 days, 120 days? Some description of these profiles is important in building the rationale for this work. E.g., how does age feature in this? Are the 9%, 26% younger with fewer or no comorbidities?

a. Unfortunately, we cannot answer this comment as we no longer hold a data sharing agreement to complete analysis of the NHFD. We believe the proportion of interest is the 74% who had outdoor mobility pre-fracture and who have not recovered this. We have better emphasised this with an update to read: ‘Despite this common failure to support 74% of patients with hip fracture to regain their pre-fracture outdoor mobility, research and clinical guidelines focus on acute rehabilitation and short-term outcomes, the focus of community rehabilitation being limited to in-home mobility(9-11)’.

3. L255 The authors state, “trained therapists” - who are/could be the trained therapists who will deliver the intervention? Please specify eg, physios, OTs, PTA, PT/OT Assistants, etc.

a. In section ‘setting’ we include the following statement: ‘Physiotherapists/occupational therapists/therapy assistants (hereafter referred to as ‘therapists’) working outside the hospital will support delivery of the OUTDOOR intervention’. We updated L255 to remove the word ‘trained’. 

4. Line (L) 253 - L257 The rationale for starting the intervention 30 days after return home is not stated. 

a. We updated to include the following sentence: ‘From 30-days to 120-days (12-weeks later) was selected given the potential to increase the proportion of patients recovering outdoor mobility in this time frame (with usual care an increase reported from 9% at 30-days to 26% at 120-days (7)(8)).’

5. Also, to clarify, the duration of the intervention will be a maximum of 6 visits within a maximum of 12 weeks unless the end-goal is achieved before then. Is this correct?

If so, please clarify this; it’s a bit confusing.

a. We updated the sentence to read: ‘The intervention will start approximately 30 days after a participant returns home and end when they have received a maximum of 6 visits within a maximum of 12 weeks, unless their mobility end goal is achieved before then’.

6. “Sixty-years of age was selected to align with the National Hip Fracture Databases definition of the target population(20).” What is National Hip Fracture Databases rationale for selecting 60 years as the target population age? How do risk factors for falls change with advancing age? And if 60 years is an important for comparison purposes, how is this likely (or not) to bias the results. These issues have not been adequately addressed in the protocol.

a. We updated the sentence to read: Sixty-years of age was selected to align with the National Hip Fracture Databases definition of the target population of older people who incur fragile hip fracture (average age at hip fracture is 83 years for women and 84 years for men(11)) (20).

7. There’s inconsistency in the spelling of “accelerometry”

a. Updated. 

8. Semi-structured interview and focus group guides were not provided.

a. Topic guides have been included as a supporting information file 4.

---

## [Editor Report · Decision Letter 1]

25 Jun 2024

Protocol for a Feasibility Randomised Controlled Trial of the ‘Outdoor’ Mobility Intervention for Older Adults after Hip Fracture

PONE-D-24-06214R1

Dear Dr. Sheehan,

We’re pleased to inform you that your manuscript has been judged scientifically suitable for publication and will be formally accepted for publication once it meets all outstanding technical requirements.

Kind regards,

Diphale Joyce Mothabeng, PhD

Academic Editor

PLOS ONE

Additional Editor Comments (optional):

Thank you for addressing reviewer comments with diligence. The journal will revert back to you about the next steps regarding publication.
---

## [Editor Report · Acceptance letter]

1 Aug 2024

PONE-D-24-06214R1 

PLOS ONE

Dear Dr. Sheehan, 

I'm pleased to inform you that your manuscript has been deemed suitable for publication in PLOS ONE. Congratulations! Your manuscript is now being handed over to our production team.

Kind regards, 

on behalf of

Dr. Diphale Joyce Mothabeng 

Academic Editor

PLOS ONE